

# A new search-and-rescue service in the Mediterranean Sea: a demonstration of the operational capability and an evaluation of its performance using real case scenarios

G. Coppini[1], E. Jansen[1], G. Turrisi[1], S. Creti[1], E. Y. Shchekinova[1,2], N. Pinardi[1,3], R. Lecci[1], I. Carluccio[1], Y.V. Kumkar[1], A. D'Anca[1], G. Mannarini[1], S. Martinelli[1], P. Marra[4], T. Capodiferro[5], T. Gismondi[5]

[1] CMCC, Fondazione Centro Euro-Mediterraneo sui Cambiamenti Climatici, Lecce, 73100, Italy
[2] SRCSB, Stuttgart Research Center Systems Biology
[3] UNIBO, University of Bologna, Bologna, viale Berti Pichat, 6/2, 40127, Italy
[4] LINKS S.p.a., Links Management and Technology, via R. Scotellaro, 73100 Lecce
[5] ITCG, Italian Coast Guard, Direzione Marittima di Bari

*Correspondence to*: G. Coppini (giovanni.coppini@cmcc.it)

**Abstract.** A new web-based and mobile Decision Support System (DSS) for Search-And-Rescue (SAR) at sea is presented, and its performance is evaluated using real case scenarios. The system, named OCEAN-SAR, is accessible via the website www.OCEAN-SAR.com. In addition to the website,
dedicated applications for iOS and Android have been created to optimise the user experience on mobile devices. OCEAN-SAR simulates drifting objects at sea, using as input ocean currents and wind data provided, respectively, by the CMEMS and ECMWF. The modelling of the drifting objects is based on the Leeway model, which parameterises the wind drag of an object using a series of coefficients. These coefficients have been measured in field experiments for different types of objects, ranging from a
person in the water to a coastal freighter adrift. OCEAN-SAR provides the user with an intuitive interface to run simulations and to visualise their results using Google Maps. The performance of the service is evaluated by comparing simulations to data from the Italian Coast Guard pertaining to actual incidents in the Mediterranean Sea.

## 1 Introduction

The Mediterranean Sea faces thousands of accidents and consequent search and rescue operations. Accidents are related to tourism and recreational activities along the Mediterranean coastlines, maritime commercial, transport and fishery operations and to the immigrations from Africa and south-eastern



Mediterranean countries. The Italian Coast Guards and Italian Navy intervened in thousands of SAR operations in national Italian and international waters.

The aim of OCEAN-SAR is to provide support to maritime authorities and operational centres (i.e the Coast Guards) to optimise the strategy of search by reducing the extent of search area extension and
increasing the probability of success of a rescue operation in case of a sea accident.  In addition to maritime authorities also private users are interested to get support in SAR activities.

Two main categories of users can be identified: 1) The public authorities responsible at different levels for SAR emergencies management such as the Coast Guard; 2) Private users (i.e. maritime transport companies, yachters) that would like to start search and rescue of an object/person lost at sea from their
boat. Private users might be even interested to search objects that the public authorities would not be responsible to search.

The SAR modelling tools are based on lagrangian modelling (Breivik and Allen, 2008) forced by wind and currents environmental fields.

Advances in high-resolution ocean operational forecast (Pinardi et al. 2003; Oddo et al. 2006; Tonani et
al. 2008) for the Mediterranean Sea, nowadays delivered by the Copernicus Marine Environmental Monitoring Service (CMEMS) Mediterranean Monitoring and Forecasting Centre  (MED-MFC), are available at the Mediterranean level providing accurate hourly forecasts also of objects in this area. The CMEMS MED-MFC modelling tools also include data assimilation system (Dobricic et al. 2004; Dobricic and Pinardi 2008) correcting the model results with observations (e.g. Sea Level Anomaly,
Temperature and salinity profiles).

**2 The Leeway model description**

The leeway model (Breivik and Allen, 2008) uses an ensemble of drifting particles to model the movement of an object due to surface currents and wind. The motion of a particle $n$ in this ensemble is
described by the model equation:

$$d\vec{x} = \left[\vec{C}(\vec{x}, t) + \vec{L}(\vec{x}, t)\right]dt \tag{1}$$

with $\vec{C}(\vec{x}, t)$ the surface current velocity and $\vec{L}(\vec{x}, t)$ the wind-induced velocity or leeway. The latter is
separated in a downwind component $(L_d)$ and crosswind component $(L_c)$ as:

$$\vec{L}(\vec{x}, t) = \begin{pmatrix} cos\theta_W & -sin\theta_W \\ sin\theta_W & cos\theta_W \end{pmatrix} \begin{pmatrix} \Omega_n L_c(\vec{x}, t) \\ L_d(\vec{x}, t) \end{pmatrix}$$

$$\tag{2}$$





Here $\Omega_n = \pm 1$ is the intrinsic orientation of the particle and $\theta_W$ the direction of the wind defined in an anti-clockwise direction with respect to the north. The components of the leeway $L_i$ ($i = c, d$ are parameterised as:

$$L_i(\vec{x}, t) = \left(a_i + \frac{\epsilon_{i,n}}{20}\right)\left|\vec{W}(\vec{x}, t)\right| + b_i + \frac{\epsilon_{i,n}}{2}$$
(3)

with $\vec{W}(\vec{x}, t)$ the wind at 10 m height, $a_i$ and $b_i$ the so-called leeway-coefficients and $\epsilon_{i,n}$ a per-particle perturbation. This perturbation is a random number drawn from a normal distribution with standard
deviation $\sigma_i$ and it represents the uncertainty on the $a_i$ and $b_i$ parameters. The result of this parameterisation is that the drift characteristics of an object can be described by the 6 object class parameters[1] $a_d, b_d, \sigma_d, a_c, b_c$ and $\sigma_c$.
The values of the object class parameters have been determined experimentally (Allen and Plourde, 1999; Allen, 2005 for various types of objects, ranging from persons in various conditions to life-rafts
and medium-size ships. The current implementation of the model defines a total of 64 classes and sub-classes.
Uncertainties on the wind and the ocean currents can be incorporated into the simulation by introducing an additional perturbation to the forcing fields. The uncertainty on the current forcing field is considered to be negligible compared to the uncertainty on the wind forcing, therefore only the perturbation for the
wind is taken into account:

$$\vec{W}(\vec{x}, t) \rightarrow \vec{W}(\vec{x}, t) + \vec{\epsilon}_{W,n}$$
(4)

The components of the perturbation $\vec{\epsilon}_{W,n}$ are random variables drawn from a normal distribution, with
standard deviation equal to the wind field uncertainty $\sigma_W$. At present no spatial or temporal correlations are taken into account for $\sigma_W$. New values for $\vec{\epsilon}_{W,n}$ are calculated for every particle $n$ at every model timestep.
The model is initialised by creating an ensemble of $N = 500$ particles of a given object class. Their initial positions are distributed according to a two-dimensional normal distribution centred at the last
known position (LKP) and with a standard deviation corresponding to the radius of uncertainty. This

---

[1] For a small number of classes the crosswind coefficients have an additional dependency on the left/right of downwind orientation $\Omega_n$. In that case the crosswind coefficients become:

$$(a_c, b_c, \sigma_c) = \begin{cases} (a_l, b_l, c_l), & \Omega = -1 \\ (a_r, b_r, c_r), & \Omega = +1 \end{cases}$$

bringing the total number of object class parameters to 9.



radius is typically of the order of a few kilometres. The initial orientation of the particles, $\Omega_n$, is chosen randomly.

To obtain the particle trajectories and the final distribution, Eq. (1) is integrated from the initial to the final time using the midpoint method with a fixed-size timestep of 360 s. The possibility of particles

changing their orientation, the so-called jibing, is taken into account by randomly changing the sign of $\Omega_n$ after every timestep. The probability of a particle changing orientation is set to 4 % per hour (Allen, 2005), i.e. a particle has a 4/1000 chance of changing

orientation at a given timestep.

### 3 Environmental forcing

The Leeway model in OCEAN-SAR system is forced by ocean currents data CMEMS MED-MFC and atmospheric wind data from ECMWF: these datasets are used for the calculation of the particles trajectories.

### 3.1 CMEMS MED-MFC products

The OCEAN-SAR uses the ocean currents analysis and forecast from the CMEMS MED-MFC

operational implementation of the NEMO model in the Mediterranean Sea (Tonani et al., 2008, Oddo et al., 2009, Tonani et al., 2009). The employed fields are currents in meridional and zonal direction at multiple depth level. The name of the CMEMS MED-MFC product used is MEDSEA_ANALYSIS_FORECAST_PHY_006_001. Currents are provided with hourly resolution on a 1/16 degree (i.e., 6 kilometers in the meridional direction) mesh on 72 vertical levels. Three type of

products are employed:

1)  The 10-days forecasts produced every day originating from the 12:00 UTC analysis/simulation

2)  The 1-week analysis produced every Tuesday from the 12:00 UTC analysis. Analyses include data assimilation (Dobricic & Pinardi 2008, Dobricic et al., 2007)

3)  The 1-day simulations produced every day (except Tuesday) from the 12:00 UTC analysis/simulation, do not include data assimilation.

### 3.2 ECMWF products

The wind forcing at 10 meters height is provided by the IFS model operated by the ECMWF. The model outputs are available with 3-hourly resolution for the first 3 days after the analysis, the horizontal

resolution is 1/8 degree (7.5 miles in the meridional direction), and the forecasts refer to the 12:00 UTC analysis.



### 4. The OCEAN-SAR operational system

The core of OCEAN-SAR runs on the CMCC super-computer system and specifically on the part called Okeanos dedicated to the maritime related services of CMCC. The OCEAN-SAR system is running in operational mode since June 2014. The System is available on web and mobile application and has been
5 configured so that registered users can submit the jobs. The service is freely available on the web and free of charge.

The OCEAN-SAR logical architecture is shown in Figure 1 and consists of the following components:

- User Interfaces (UIs) for the web and mobile client devices;
10 - Data pre-processing system;
- Sea Situational Awareness (SSA) platform (web portal, map service and message broker);
- The Complex Data Analysis Module (CDAM) in the computing centre.

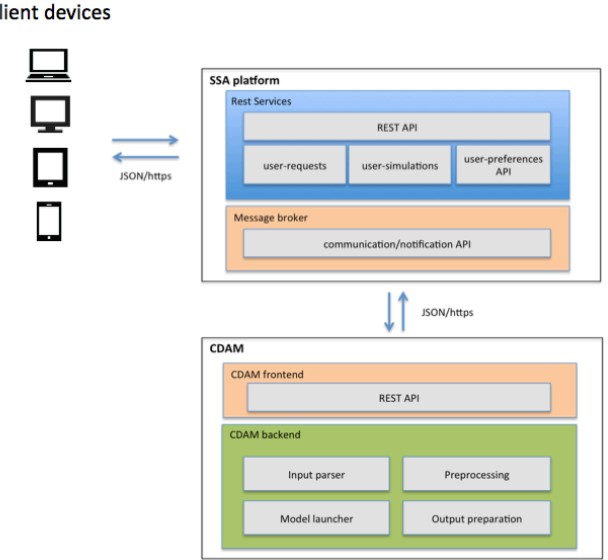

**Figure 1: OCEAN-SAR high-level architecture**

Only the relevant components of the system will be described in this paper, for a comprehensive description of the SSA platform, CDAM and their components see Mannarini et al. (2016).





All the messages are exchanged by the components using the JavaScript Object Notation (JSON) format. The communication and the data exchange between the client devices and the SSA platform via the Representational State Transfer (REST) web services established by means of the message broker component that receives and forwards the requests to the CDAM hosted on the computational cluster

Okeanos at CMCC (Figure 2).

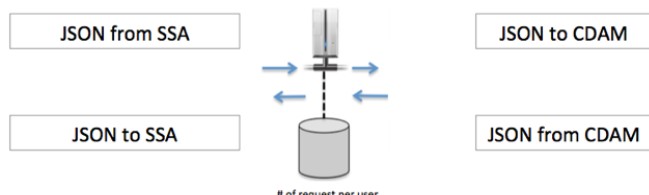

**Figure 2: Message Broker**

The OCEAN-SAR pre-processing shell script parses the input JSON string in order to obtain the values of parameters required by the model to run the simulation. The parameters names and default values are listed in Table 3 of Appendix 1. The system, before running the simulation, checks the data availability for the defined domain and simulation period. The wind data are used only if the 'depth' argument is '1', for the sub-surface simulations no wind is used. According to the 'simulation time' argument the

current and wind data are selected. The NetCDF files are extracted into a specific location of the file system where a folder is created for each simulation. The depth slab at a required depth is made by using NetCDF Operators (NCO) utilities. The data concatenation is performed across the time variable. The Fortran interpolation procedure called SeaOverLand is carried for the extrapolate data near to the coast. Similar pre-processing procedure is completed for the wind by the Data Pre-processing system.

Then the file is created dynamically and fetched to the lwseed executable code that creates initial positions of an ensemble of drifters in the leeway input file. For every step an error management procedure is implemented that for the moment being will kill the job ad give the message to the UI. After the Leeway model runs successfully, the output data are coded in plain text ASCII format and written in the output directory corresponding to the number of the simulation user request. The system

checks for the size of the output data and subsequently, if it is not empty, stores the output in JSON format as a new file.

The communication via REST web services is established through the Message Broker, a component of the SSA platform that receives the requests and forwards them to the CDAM hosted on the computational cluster Okeanos. On Okeanos the model is executed and returns the results to the user via

the message broker.

All the user requests as well as the results of the simulations are stored in a database by a component of the SSA platform, which ensure the data persistency (Figure 3).




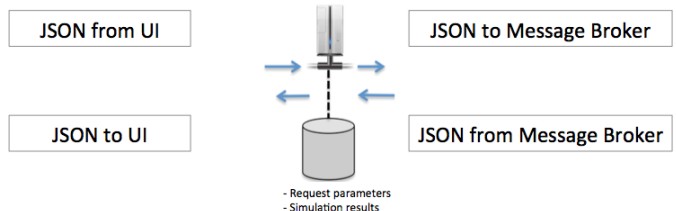

**Figure 3: SSA component responsible for data persistence**

Thanks to this component, each user can visualize at any time the results of previously performed
simulations and view the parameters used to run the simulation.
The data flow in Figure 4 shows how the messages are exchanged in JSON format among the different
components via REST web services.

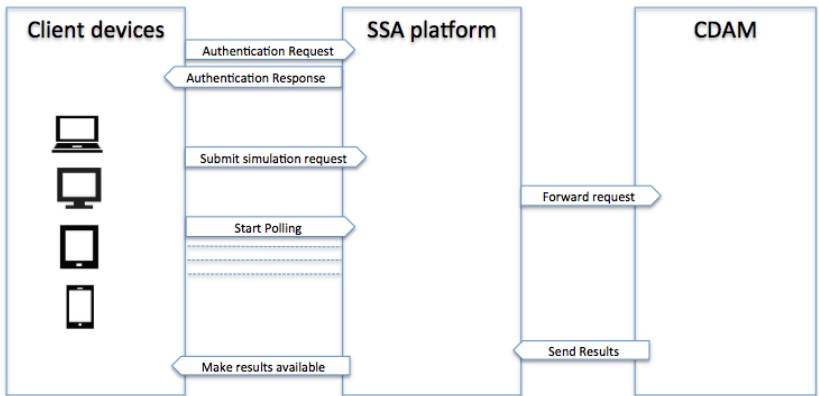

**Figure 4: Data flow among components**

On Okeanos the model is executed using Simple Linux Utility for Resource Management (SLURM)
and the output are produced in ASCII format. The system converts the ASCII output data into JSON
format in order to send back to CDAM (see Figure 2). The CDAM then sends the output to Message
broker. In the meanwhile, client applications retrieve the results by polling for their availability,
checking the content, and extracting the information required for the visualization. After completion of
the loop the results are provided to the user via graphically displayed fields.


### 4.1 Web and mobile applications

The web and mobile applications (iOS and Android) provide the access for the user through their user interfaces. The web application has been adapted for use on mobile devices (tablet and mobile phone) and app are available on the Apple and Android stores as free app.

The User Interface (UI) allows setting the parameters of the simulation, submitting requests and displaying the results of simulations on a map. The configurable parameters are shown in Figure 5. The parameters can be grouped as follows:

- General parameters (simulation name, object category);

- Last known position (start position, end position, start date, end date);

- Simulation duration;

- Forecasting system (wind and currents models);

- Display settings (environmental fields on/off);

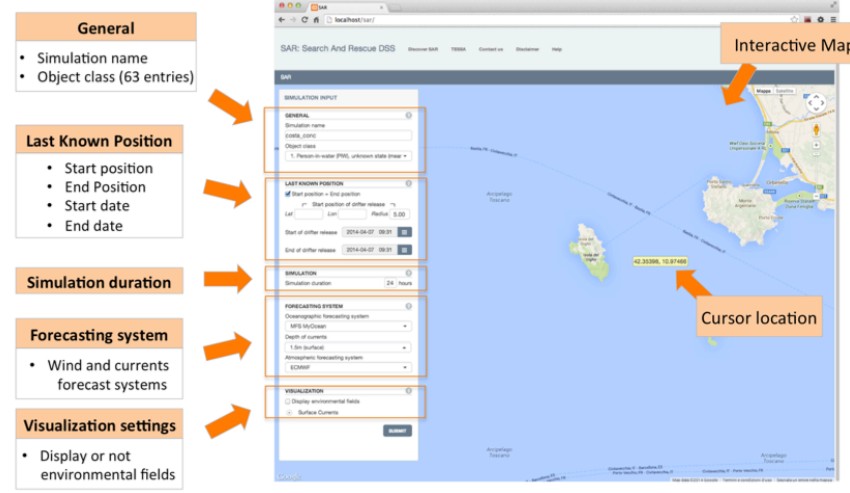

**Figure 5: Parameters of Simulation displayed on the UI**

The meaning of each set of parameters is explained (see        Figure 6) through a yellow box that appears on the UI after a click of the mouse.





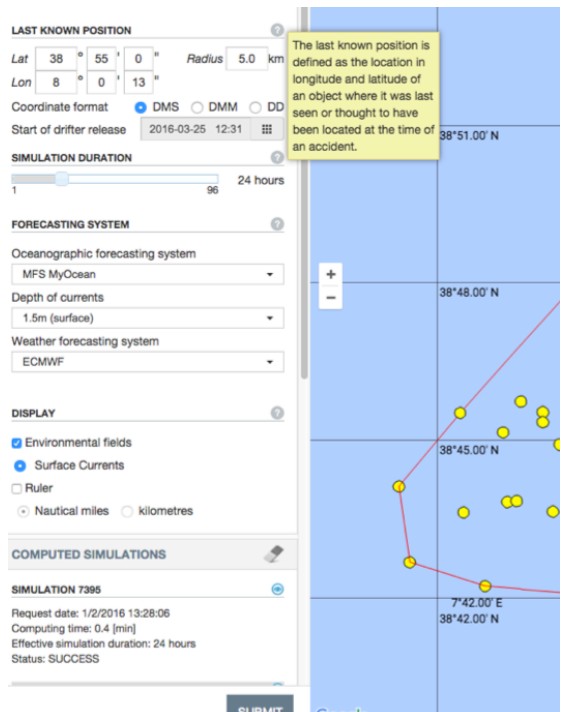

Figure 6: help option overview of parameters





The results of a simulation can be displayed in the UI in two ways: with or without the magnitude of environmental fields (**Figure** 7).

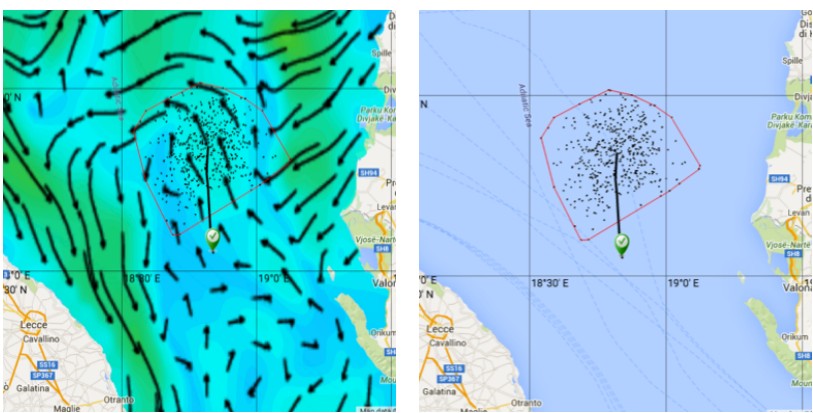

**Figure** 7: display of the simulation results with environmental fields (left) and without environmental fields (right).

The UI (Figure 8) visualizes the following output/results of the model:

- Drifters final positions (yellow or red markers);

- Mean trajectory drift (black curve);

- Drifters stranded along the coast line (red markers).



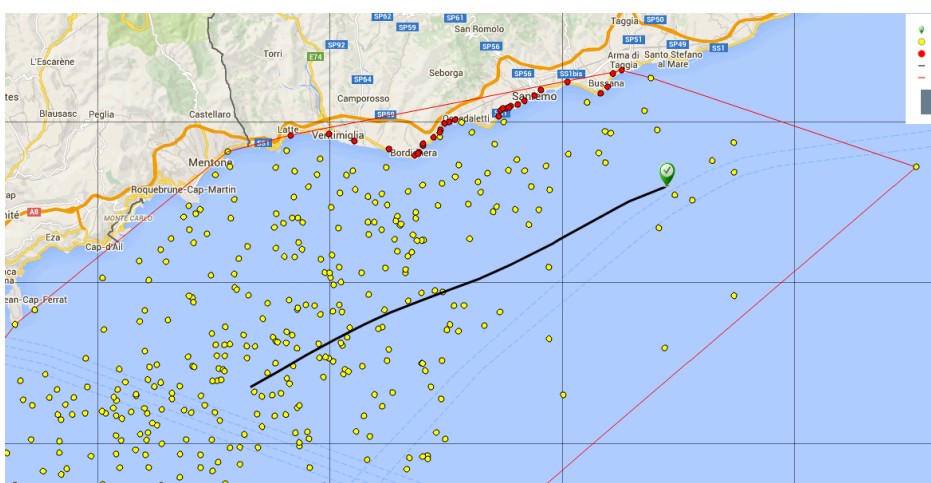

Figure 8: detailed visualization of the output of the model

**5 Real case scenarios**

OCEAN-SAR has been validated by reconstructing real case scenarios (accidents and field exercises) using information provided by the Italian Coast Guard. A selection of these real-case scenarios is presented in this section.

The Italian Coast Guard provided the information of 6 past events (Table 1) with different
characteristics:
   1)  Field exercise in Reggio Calabria waters:

   a.  Search and rescue dummy (here called Calabria1)

   b.  Raft (here called Calabria2)

For the raft used in this field exercise there was not an exact correspondent type of object in the IAMSAR manual (IMO, 2016) and therefore, following the indications by the Italian Coast Guard, a "shallow ballast with no drogue" raft was used for the simulation. For the dummy the type of object called "Person in the water (PIW), unknown state (mean value)" was used.

2)  Commercial fishing vessel used as a migrant ship





In this case the commercial fishing vessel with on board many migrants was simulated using the type of object "Commercial fishing vessel (14-30m) Troller" which was the correspondent object in the IAMSAR manual. The boat was found by the Italian Coast Guard at 00:40 UTC of the day 1 December 2013 (LAT 38°08' N- LON 018°21'E), the sea state was rough (more than 6m of waves height) and the

the boat was drifting with failed engine, there was no possibility of immediate rescue and the ship was reached one day later when the sea state was better and allowed the transhipping of the migrants. The transhipment started at 12:05 UTC on 2 December 2013 at the position 38°39'6"N, 018°21'E after 36 hours of drift.

3) 60 m long ship used as migrant ship

A ship 60 meters long was transporting immigrants who were rescue and transhipped. After the immigrant transhipment it was not possible to tow the ship, therefore it was left drifting. The position was recorded through successive sightings and this data is used for the validation of OCEAN-SAR. The length of the ship was approximately 60 meters, and there is not a specific correspondent type of object

in the IAMSAR manual. The most similar ship type is the coastal freighter, which is used in the simulation of this paper. The time and positions recorded for the ship are reported in table 1 (Migrantship2).

4) Small boat

On the 11th of December 2014 at 18.00 UTC a person on board of a small boat started drifting from the position 40°00'55"N 018°26'10"E. The location and time have been reconstructed based on the information provided by the person on board and are expected to be quite accurate (error estimated to be less than 1 nm and less than 2 hours). After 2 days, at 09:05 UTC on 13 December 2014, the person was found alive on his boat in the position 39°29'N 018°12'E. The type of object in the IAMSAR manual

that best fit with the real boat is skiff V hull smaller than 6 metres.

5) Person at sea lost from a ferry

A person fell overboard from a ferry between 20:30 on the 11 July 2013 and the 23.30 UTC. The body of the person was foung at 10:30 of the 12 July 2013 at the position LAT 39° 54'.716 N    LON 010°

06'.294 E. In this case the last known position follows the AIS path corresponding to the positions of the ship during the time the person might have fallen overboard.





| Name | Class | Seeding start | | | | Seeding end | | | | Recovery | | |
|---|---|---|---|---|---|---|---|---|---|---|---|---|
| | | Time | Latitude | Longitude | Radius | Time | Latitude | Longitude | Radius | Time | Latitude | Longitude |
| calabria1 | 7 | 2014-11-13 22:00 | 37°45'N | 15°36'E | 0,1 | | | | | 2014-11-14 11:01 | 37°41.6'N | 15°31.7'E |
| Calabria2 | 1 | 2014-11-13 22:00 | 37°45'N | 15°36'E | 0,1 | 2013-07-11 23:30 | 40º25.06'N | 10º33.07'E | 0,1 | 2014-11-14 10:40 | 37°42.9'N | 15°31.1'E |
| migrantship1 | 43 | 2013-12-01 00:40 | 38°08'N | 18°21'E | 0,1 | | | | | 2013-12-02 12:05 | 38°39'6''N | 18°21'E |
| Migrantship2 | 49 | 2014-12-05 16:08 | 36°25'N | 18°33'E | 0,1 | | | | | 2014-12-08 10:57 | 36°49'N | 19°39'E |
| Migrantship2 | | | | | | | | | | 2014-12-05 21:00 | 36°31'N | 18°36'E |
| Migrantship2 | | | | | | | | | | 2014-12-06 12:40 | 36°42'N | 18°53'E |
| Migrantship2 | | | | | | | | | | 2014-12-06 17:00 | 36°43'N | 19°02'E |
| Small boat | 39 | 2014-12-11 18:00 | 40°00'55''N | 18º26'10''E | 0,1 | 2014-12-11 22:00 | 40°00'55''N | 18º26'10''E | 0,1 | 2014-12-13 09:05 | 39º29'N | 18º12'E |
| Ferry | 1 | 2013-07-11 20:30 | 39°33.16'N | 9°58.02'E | 0,1 | 2013-07-11 23:30 | 40º25.06'N | 10º33.07'E | 0,1 | 2013-07-12 10:30 | 39°54.716'N | 10º06.294'E |

Table 1 Real case scenarios reconstruction information. All times are UTC.





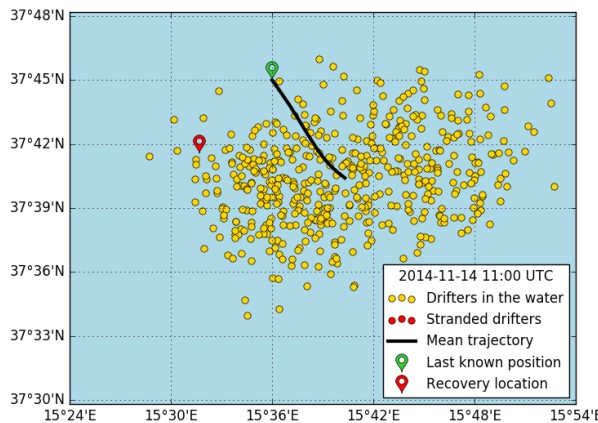

**Figure 9: results of the "Calabria1" simulation done using OCEAN-SAR showing the positions (yellow points) of the particles at the end of the simulation 11:00 UTC of 14 November 2014. The red place-mark indicates the position in which the dummy was collected. The green place-mark represent the position in which the dummy was launched at sea and started drifting. The black line shows the mean trajectory of the simulated particles.**

In the reconstruction of the Calabria1 (Figure 9) case the final position of the simulated drifters shows a partial agreement with the observation: the dummy drifted south west and its final position after 13 hours at sea is contained in the area identified by the OCEAN-SAR system but the simulation shows an area more shifted eastward.





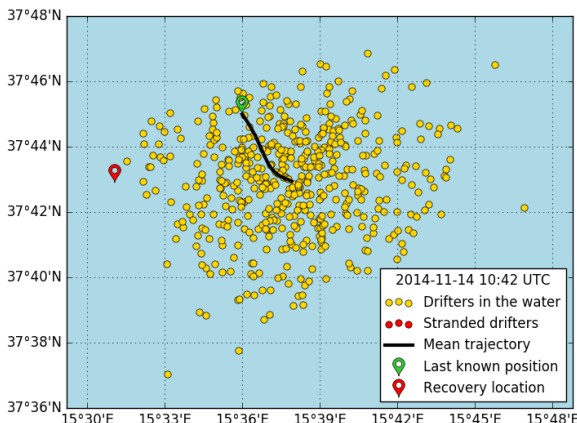

**Figure 10: results of the "Calabria2" simulation done using OCEAN-SAR showing the positions (yellow points) of the particles at the end of the simulation 10:40 UTC of 14 November 2014. The red place-mark indicates the position in which the raft was collected. The green place-mark represent the position in which the dummy was launch at sea and start drifting. The black line shows the mean trajectory of the launched particles.**

In the reconstruction of the Calabria2 (Figure 10) case the final position of the simulated drifters shows a partial agreement with the observation: the raft drifted westward and its final position after 13 hours at sea is just outside the area identified by OCEAN-SAR system while the simulation shows an area more shifted eastward. OCEAN-SAR system does not show appreciable differences between the search and rescue dummy simulated in Calabria1 and the raft simulated in Calabria2. In reality the raft moved westward faster than the dummy.





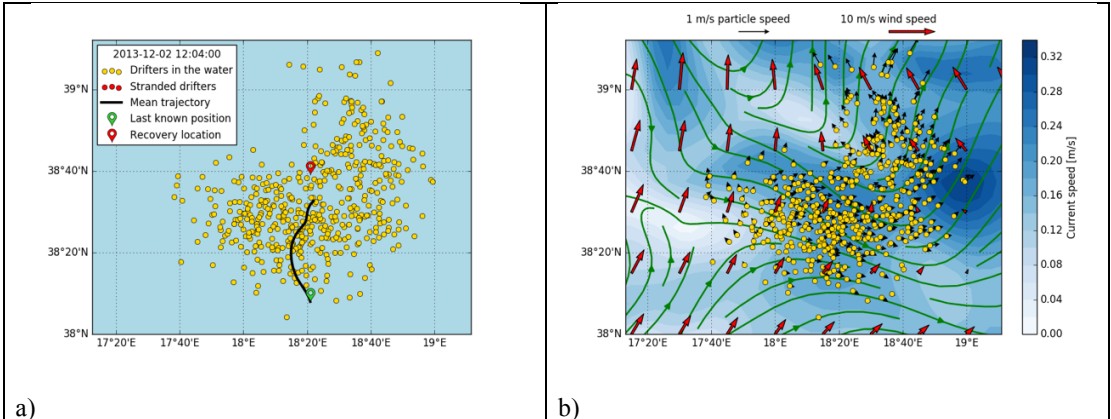

a)                  b)

**Figure 11: a) results of the "migrantship1" simulation done using OCEAN-SAR showing the positions (yellow points) of the particles at the end of the simulation 12:05 UTC of 2nd December 2013. The red place-mark indicates the position in which the ship was found. The green place-mark represent the position in which the ship was found and start drifting. The black line shows the mean trajectory of the launched particles; b) for each position of the drifters (yellow points) little black arrows indicates the direction and speed of their movement. Green streamlines indicate current direction and colored background shows their magnitude (white=low, blue=high). The wind field is shown by the red vectors.**

In the reconstruction of the migrantship1 case (Figure 11) the final position of the simulated drifters is in good agreement with the observations: the ship drifted northward and its final position after approximately 12 hours is inside the area identified by the OCEAN-SAR system. Figure 11b presents the fields of wind and currents and the mean velocity of for each of the particles.





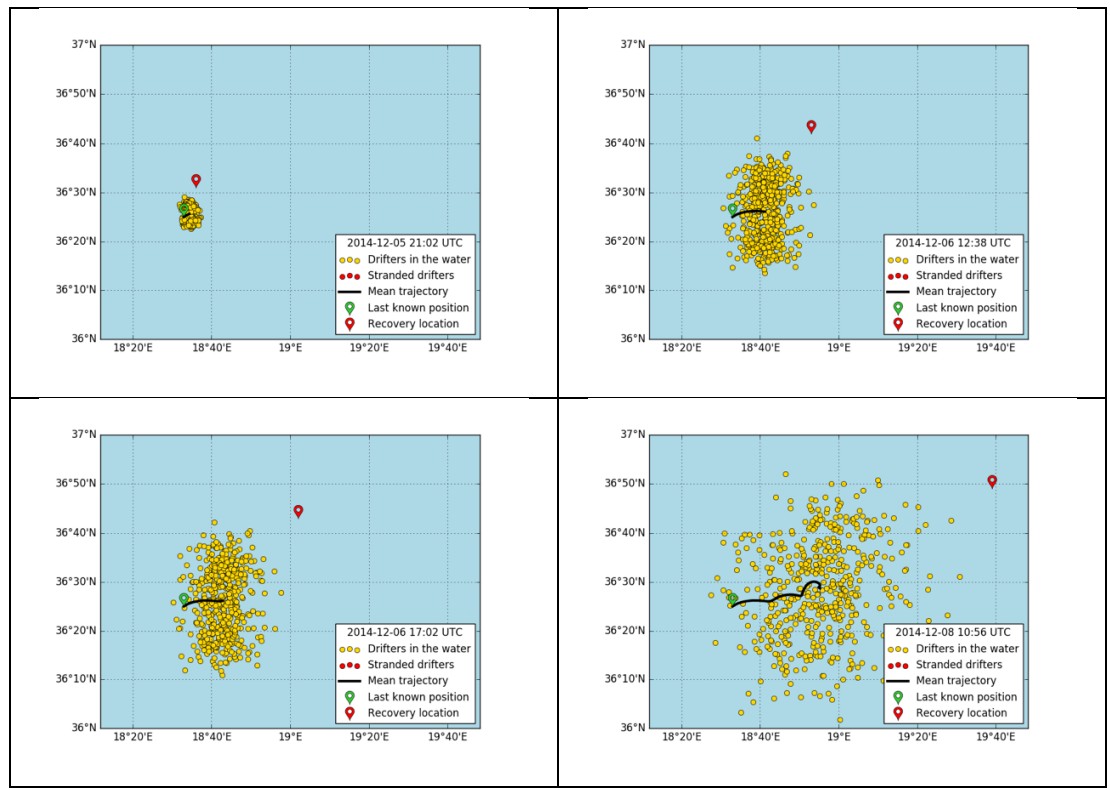

**Figure 12: results of the "migrantship2" simulation done using OCEAN-SAR showing the positions (yellow points) of the particles at the end of the simulation 10:57 UTC of 8th December 2014. The red place-mark indicates the position in which the ship was rescued. The green place-mark represents the position in which the ship was found but when it was not possible to rescue. The black line shows the mean trajectory of the launched particles.**

In the reconstruction of the migrantship2 (Figure 12) case the area identified by the simulated drifters does not include the final position of the ship, which drifted for almost 3 days (66 hours). The simulated drifters are moving towards the east, while the ship is faster and moves further north-westward.



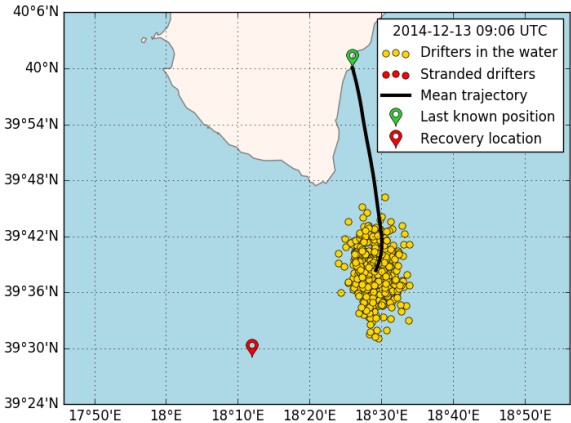

**Figure 13: results of the "Small boat" simulation done using OCEAN-SAR showing the positions (yellow points) of the particles at the end of the simulation 09:05 UTC of 13th December 2014. The red place-mark indicates the position in which the boat was rescued. The green place-mark represents the last known position. The black line shows the mean trajectory of the simulated particles.**

In the reconstruction of the Small boat case (Figure 13) the final position of the simulated drifters is not in agreement with the observation: the small boat drifted more south west and its final position after 15 hours at sea is outside the area identified by OCEAN-SAR system. OCEAN-SAR system predicted correctly a southward movement of the boat, but in reality the boat moved faster further westward. The westward position of the boat can also be explained by the fact that the person on the boat was paddling towards land and the Coast Guard explain that this was resulting on a westward movement of the boat.





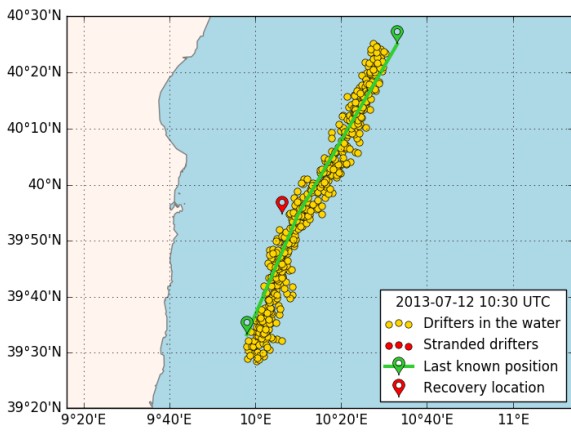

**Figure 14: results of the 'Ferry' simulation done using OCEAN-SAR showing the positions (yellow points) of the particles at the end of the simulation 10:30 UTC of 7th December 2012. The red place-mark indicates the position in which the person was found at sea after 14 hours from the last time he was seen on board of the ferry. The green segments represent the route of the ferry along which the particles have been simulated.**

In the reconstruction of the 'Ferry' case (Figure 14) the final position of the simulated drifters is in partial agreement with the observed location of the body found at sea after 13 hours of drifting. The real position is few hundred meters westward from the OCEAN-SAR area.

10  OCEAN-SAR was used already several time by Italian Coast Guards in real case actions of emergency related to SAR. In this paper we present the case of a diver lost in Monopoli, which body was found few hours later (Figure 15). The OCEAN-SAR simulation results are in good agreement with the real drift of the body.



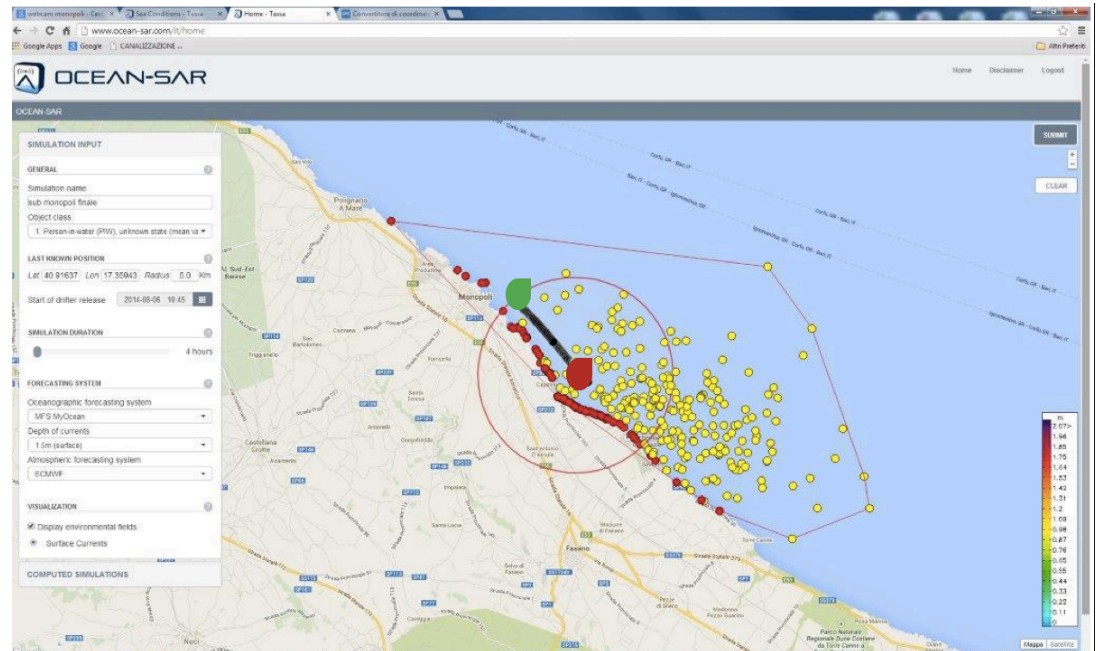

**Figure 15: results of the real case scenario called MONOPOLI, simulation done using OCEAN-SAR showing the positions (yellow points and red points on the coast) of the particles at the end of the simulation 14:45 UTC of 6[th] August 2014. The large red place-mark indicates the position in which the person was found at sea after 4 hours. The green point represents the last known position at which the diver was lost at 10:45 UTC of the 6[th] August 2014.**

**6. Conclusions**

OCEAN-SAR is an operational decision support system, based on the Leeway model and implemented as a complete system for on-demand simulation of drifting objects at sea. It uses as input ocean currents and wind data forecasts from third party providers that are dynamically prepared. The DSS is freely available on the web and for mobile devices.

The interaction with the users and the participation of authors from Italian Coast Guard to the co-design and testing of the OCEAN-SAR allowed the customization of the service towards the users' needs and requirements.

The experience of OCEAN-SAR highlights that research and development activities, in the our case ocean and Lagrangian modelling, should as much as possible aim to converge towards operational applications so that their results and finding are tested in operational mode and in real case scenarios by



the users, often situations in which weak components of the systems and bugs are highlighted. The operational testing also helps to demonstrate the effectiveness and importance of research results for supporting societal challenges and in our case the benefit for maritime safety and for saving life at sea. As possible future improvements to OCEAN-SAR we foresee the use of the high-resolution sub-

regional and coastal operational models for currents and wind forecasts, especially in coastal areas, and the implementation of an Application Programming Interface (API) layer which will allow users to interface their own software tools with OCEAN-SAR, the further testing by users and the integration with users' software already existing and in use for the managing of SAR emergency at sea.

**Acknowledgments**

This work was performed in the framework of the TESSA Project PON01_02823 supported by PON Ricerca & Competitività 2007-2013 cofunded by UE (Fondo Europeo di sviluppo regionale), MIUR (Ministero Italiano dell'Università e della Ricerca), and MSE (Ministero dello Sviluppo Economico). We thank Copernicus Marine Environment Monitoring Service - CMEMS MED-MFC for the provision

of the ocean forecasting products, Servizio Meteorologico Aeronautica Militare Italiano for the provision of the wind products and Italian Coast Guard, in the person of C.F. Sirio Faè, for the kind cooperation, important suggestions and for the provision of the data related to the real case scenarios used for testing OCEAN-SAR in this paper.

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

**Appendix 1:  Description of user –defined parameters for the SAR DSS**

15   To start the Leeway simulation a string of input arguments from the CDAM is created that contains the full set of user-defined parameters introduced via the web UI.  The set of parameters is listed in Table 2

| # | Name | Description | Format: Default values | Units |
|---|------|-------------|------------------------|-------|
| 1 | id | Query Unique identifier | Integer number: > 0 | - |
| 2 | simulation_name | Name of the simulation | Char: no default value | - |
| 3 | object_class | Class of the object in water | Integer number: 1 | - |
| 4 | start_lat | Start position, latitudes | Float: no default value | Degrees |
| 5 | start_lon | Start position, longitudes | Float: no default value | Degrees |
| 6 | start_rad | Start radius | Float: 5.00 | Meters |
| 7 | start_time | Start time of simulation/seeding | ISO-8601: 2014-04-14T10=00Z; | - |
| 8 | simulation_time | Duration of simulation | Integer: 24 | Hours |
| 9 | depth | Depth of the currents | Integer: 1 | No unit/model level number |
| 10 | slnms=01.04; | Start longitude in minutes and seconds | Float: 01.04 | Minutes-seconds |
| 11 | eltd=42; | End latitude in degrees | Integer: 42 | Degrees |
| 12 | eltms=18.49; | End latitude in minutes and seconds | Float: 18.49 | Minutes-seconds |
| 13 | elnd=11; | End longitude in degrees | Integer: 11 | Degrees |

**Table 3 Description of  user-defined parameters. The data are sorted according to Name of parameter as written in the input**
20   **string, Description, Value type and Units.**