# Peer review of "A new search-and-rescue service in the Mediterranean Sea: a demonstration of the operational capability and an evaluation of its performance using real case scenarios"

_Natural Hazards and Earth System Sciences, 2016_

## Referee Comment (RC1) · A. Allen (Referee) · 16 Jun 2016

Font Size: Sections and Figure captions should have larger font sizes.

page 1, Line 5-10, authors' superscripts (1-5) don't match notes: 1,5,3,4,4 of authors' institutions. page 4, lines 15-20; The discussion of the ocean currents, is a bit confusing, as to exactly which ocean currents are used by the OCEAN-SAR. I assume that is due to the production cycle of the model (see http://medforecast.bo.ingv.it/), which rotates throughout the week. Also, used of the term 'employed', is unclear, suggest 'used by' or 'accessed by OCEAN-SAR' Page 6, lines 18- 20; this discussion on SeaOver-

Land and lwseed should be a higher level, instead of referring to these routines or sub modules, state what is accomplished with these routines. SeaOverLand extrapolates data near the coast. What data? sea currents, particles, wind? 'lwseed' is mentioned here and only here, no real need for it at all. Perhaps something along the lines of "The initial positions are randomly generated for LKP before passing to the drift trajectory module of OCEAN-SAR"

page 6 lines 21-22: suggest the following: 'For every step an error management procedure is implemented that may cause the processing to stop, killing the job, and will post an error message with details to the UI.'

page 8, line 9, "LKP(start position, end position, start date, end date)' in the present online version of OCEAN-SAR only LKP (start position and start time) are available. Do the authors have a different version of OCEAN-SAR? If so, perhaps, the paper should include a reference to the version (or date) of OCEAN-SAR they used.

Section 5 Real Case Scenarios Figures or photos of any of the search objects used in the case studies would be useful.

Table 1: Calabria#1 (SAR dummy) is Class 7, when this should have been Class 1, or my recommendation Class 6 PIW deceased. Calabria#2 (raft), switched Class1 with Calabria#1. Should be Class 7. Why is there a Seeding End Time and Position for this? Where both the SAR dummy and the raft deployed at the same time and location? MigrantShip#1, FV Japanese side-stern trawler (#45) (from Suzuki and Sato (1977), was a 62 m vessel, similar in length to Migrant#2 at 60m.

In general, using actual SAR cases are of limited value in validating a SAR trajectory model. Either we have good agreement, or not. If not, then the question are: was LKP correct? Was the correct or most appropriate search object used (are the leeway equations right)? What are the uncertainties in the winds? What are the uncertainties in the currents? The authors should al least recognize that these uncertainties exist.

---

## Referee Comment (RC2) · K.-F. Dagestad (Referee) · 19 Jul 2016

General comments:

The paper describes the interactive search-and-rescue service available on http://www.ocean-sar.com. The web interface works quite nicely, and is fairly well described in the paper. However, there are several details in the description which should be fixed and improved.

Specific comments:

[Figure]

page 1, line 30 and page 2, line 1: "thousands of rescue operations" are mentioned, but not the time span over which they occurred. Per year? Per decade? More specific numbers of the rate (e.g. incidents per year) would be welcome.

page 3, line 18: uncertainty in current forcing is considered to be negligible compared to uncertainty in wind forcing. This should be justified somehow.

page 4, line 29: Is ECMWF model only available for 3 days ahead? Web interface (present version) allows simulations of duration up to 97 hours (4 days).

page 4, line 30: resolution should be given in km (as for currents above), not miles.

page 5, line 2: Does details such as the name of the server component "Okeanos" have any meaning to people outside of CMCC? Should otherwise be omitted, rather referring to "the server".

page 5, line 9 onwards: The listing of components does not match directly the diagram in Figure 1. Where (and what) is the "Data pre-processing system"? Are the UIs the same as the "Client devices"? Is the server "Okeanos" (where the model is actually run?) a part of CDAM or outside? Later (page 7, line 11) it is said that output from Okeanos is "sent back to CDAM", so apparently it is another machine/server?

page 6 line 5: It is not clear what Figures 2 and 3 add to the paper. The figures should be made (or described) clearer, or perhaps omitted.

page 6, line 12: The term "The system" which is used here (and also later) is quite vague. Should be more specific on which component (ref Fig 1) is discussed.

page 6, line 14: All the object categories are for objects at the surface, so why is there an option to use ocean currents at lower levels? If there is no wind, any object would (in the Leeway model) simply drift with the currents in the exact same manner.

page 6, line 18: details such as names of Fortran routines (SeaOverLand, lwseed) should not be mentioned, but rather which actions are performed by the code.

[Figure]

page 6, line 29: How can the CDAM module use the "Message broker" to return results to the SSA module, when the "Message broker" is illustrated (Fig 1) to be a component of the latter? Or does CDAM has its own "Message broker"?

page 7, line 8: Very much overlap with Figure 1, could these figures be merged?

page 7, line 15: "results are provided to the user via graphically displayed fields". Is WMS or any other functionality used here? This should be described in more detail. Also in the web interface, only currents can be shown as "environmental fields", but it would also be of interest to show the wind. It is also not clear whether the displayed current field is for the start or end time of the simulation. Ideally one would like to have a time slider in the user interface, where both the currents/winds and particles would update/move.

page 8, line 14: On Figure 5, the screenshot shows that start and end-time of LKP can be specified, but not the "end position" which is indicated in the text box outside of the figure. Several of the validation cases (Table 1) have used seeding on a line between two points, which is not possible through the shown interface. In the present version at http://www.ocean-sar.com it is also not possible to specify the end-time, only start-time and start-position.

page 10, line 6: also a red contour line is shown around the drifters, is this a convex hull?

page 11, line 5: It should be mentioned which of the current products have been used for the validation exercises. The accuracy of the current is of primary concern, and is more important for the final results than the "search-and-rescue system" itself.

page 11, line 11: The layout/indents in the following is a bit messy.

page 12: positions (longitude and latitude) are given both in Table 1 (page 13) and in the text, giving possibilities for inconsistencies. I would suggest giving coordinates only in the table. Coordinates are given sometimes as minutes plus decimal degrees (e.g.

[Figure]

37°41.6'N) and sometimes as minutes, degrees and arc-seconds (e.g. 38°39'6"N). In some cases notation is wrong: 39° 54'.716 N should probably be 39° 54.716' N. I would prefer to use only decimal degrees, but at least notation should be consistent.

page 13, table 1: units could be given in table column headers. Radius is given as "0,1", probably meaning "0.1", and this is presumably in km (and not degrees?), i.e. 100 meters? For Calabria1/2 the Leeway classes 1 and 7 have been interchanged. The wind conditions makes a real difference to the simulations, and would be a useful addition to the table or Figure captions below. E.g. "winds between 5 and 10 m/s from south-south-west.". E.g. for high winds one would expect a larger difference between a raft and a person-in-water than for weaker winds.

page 17, Figure 12: four figures are shown, presumably a time series of the evolution. This is useful, but the caption and discusses only the final position (lower right figure).

page 22, Table 3: in rows 10-13, default values are shown also in the Name-column. "Format" and "Default values" should be separate columns. Radius for seeding is stated to be in meters, but correct is presumably kilometers. Longitude and latitude are stated to be given as "minutes and seconds", but correct is presumably decimal degrees, as the default values are given as single floats.

Technical corrections:

There are several minor errors and misprints throughout the text which should be checked and corrected. There are also several badly/incompletely phrased sentences such as e.g. the caption on Figure 5: "Parameters of Simulation displayed on the UI".

---

## Author Comment (AC1) · 7 Oct 2016

The comment was uploaded in the form of a supplement:
http://www.nat-hazards-earth-syst-sci-discuss.net/nhess-2016-175/nhess-2016-175-AC1-supplement.zip

---

## Author Comment (AC2) · 7 Oct 2016

The comment was uploaded in the form of a supplement:
http://www.nat-hazards-earth-syst-sci-discuss.net/nhess-2016-175/nhess-2016-175-AC2-supplement.zip